# Preparation of Heavily Doped P-Type PbSe with High Thermoelectric Performance by the NaCl Salt-Assisted Approach

**DOI:** 10.3390/molecules28062629

**Published:** 2023-03-14

**Authors:** Xinru Ma, Xuxia Shai, Yu Ding, Jie Zheng, Jinsong Wang, Jiale Sun, Xiaorui Li, Weitao Chen, Tingting Wei, Weina Ren, Lei Gao, Shukang Deng, Chunhua Zeng

**Affiliations:** 1Faculty of Science, Institute of Physical and Engineering Science, Kunming University of Science and Technology, Kunming 650500, China; 2College of Chemistry and Materials Science, Hubei Engineering University, Xiaogan 432000, China; 3Nuode New Materials Co., Ltd., Shenzhen 518048, China; 4Education Ministry Key Laboratory of Renewable Energy Advanced Materials and Manufacturing Technology, Yunnan Normal University, Kunming 650500, China; 5Faculty of Materials Science and Engineering, Kunming University of Science and Technology, Kunming 650093, China

**Keywords:** chemical doping, salt-assisted, NaCl doping, PbSe, thermoelectric material, thermoelectric performance

## Abstract

Thermoelectric (TE) technology, which can convert scrap heat into electricity, has attracted considerable attention. However, broader applications of TE are hindered by lacking high-performance thermoelectric materials, which can be effectively progressed by regulating the carrier concentration. In this work, a series of PbSe(NaCl)_x_ (x = 3, 3.5, 4, 4.5) samples were synthesized through the NaCl salt-assisted approach with Na^+^ and Cl^−^ doped into their lattice. Both theoretical and experimental results demonstrate that manipulating the carrier concentration by adjusting the content of NaCl is conducive to upgrading the electrical transport properties of the materials. The carrier concentration elevated from 2.71 × 10^19^ cm^−3^ to 4.16 × 10^19^ cm^−3^, and the materials demonstrated a maximum power factor of 2.9 × 10^−3^ W m^−1^ K^−2^. Combined with an ultralow lattice thermal conductivity of 0.7 W m^−1^ K^−1^, a high thermoelectric figure of merit (*Z*T) with 1.26 at 690 K was attained in PbSe(NaCl)_4.5_. This study provides a guideline for chemical doping to improve the thermoelectric properties of PbSe further and promote its applications.

## 1. Introduction

Thermoelectric (TE) materials capable of transforming scrap thermal energy into electric power have attracted considerable attention [1,2]. To promote the sustainable development of the environment and economy, it is crucial to explore suitable TE materials with high TE performance for their practical application. Currently, based on the existing thermoelectric materials such as Bi_2_Te_3_ and SiGe, the corresponding conversion efficiency of them are around the value of 10%, which is far away from the demanding value of 20% [3]. Among the advanced *p*-type thermoelectric materials, lead chalcogenide has generated widespread interest due to its low thermal conductivity, high carrier fluidity, and high symmetrical structure [4,5], in which the PbTe has been widely studied. Notably, regarding the high melting point and rich content of the element, PbSe (a close analog of PbTe), with a similar two-valley valence band structure to PbTe, has also captivated extensive research, but the unsatisfactory TE properties still hinder its practical applications [6]. To obtain higher conversion efficiency, it is necessary to improve the dimensionless of merit (ZT), which is described as ZT = α^2^σ T⁄κ, where σ represents the electrical conductivity and α denotes the Seebeck coefficient. The product α^2^σ is referred to as the power factor. The total thermal conductivity is composed of the lattice thermal conductivity (*κ_L_*) and the electron thermal conductivity (*κ_e_*), and T represents the absolute temperature [7]. Generally, there are two ways to enhance the ZT values, one is optimizing carrier concentration and changing the energy band to boost the PF, and another is reducing the lattice thermal conductivity through the initiation of atomic defects and formation of nanostructures [8,9,10,11]. Excitingly, chemical doping can not only modify the electronic structure but also optimize the carrier concentration, which has been illustrated as a successful control approach for tuning the TE material properties [12,13,14]. By properly introducing dopants, the energy distance from the light band to the heavy valence band of lead chalcogenide can be greatly decreased—such “band convergence” is beneficial to improve the TE performance and especially increasing the Seebeck coefficient [15].

To tune the band structure of *p*-type PbSe, many different dopants have been examined, such as by substituting Cu/Sr/Ba/K/Ag for Pb or Te/Cl for Se [16,17,18,19]. In addition, the doping method has also been explored [20,21]. Compared with other processes, the flux method is widely considered to obtain highly crystallized and phase-pure samples [22]. Due to the slow cooling process, the stable liquid environment (flux) provides sufficient space and ample time for the constituent atoms to nucleate and self-organize [23]; it is accessible for the experiment since the atmospheric pressure and the working temperature are below the melting point [24]. In the preliminary work, both Sn and Ga have been proven as the feasible effective flux for thermoelectric materials [25,26]. Moreover, given the cost, environmental friendliness, and sustainability, NaCl has become the best choice as the flux. Excitingly, Deng’s research group has adopted NaCl as the flux to prepare the Cu_2_Se_1+x_(NaCl)_2.5_ (x = 0, 0.01, 0.02, 0.03) and β-Zn_4_Sb_3_ successfully, thus presenting excellent thermoelectric properties [27,28]. Furthermore, theoretical calculations have been widely performed to understand the effect of chemical doping on band structure and thermal transport properties of TE materials due to the predictability and reliability, providing guidance for optimizing and designing TE materials [29,30,31]. 

Thus, chemical doping combined with the suitable preparation process, the PbSe-based materials with high TE performance could be achieved by structural optimization or electronic band regulation, associating with the theoretical calculation, contributing to understanding the nature of the doping effect, and providing a substantive improvement in TE materials. Aiming toward a fundamental understanding and practical development of PbSe-based thermoelectric materials, the pursuit of highly effective TE materials toward ultimate commercialization necessitates greater knowledge of the effects and regulators of chemical doping in light of the significance of chemical doping for TE materials. Herein, we present tuning TE performance via chemical doping and processing improvement by NaCl flux, employing the Na and Cl substitution as a prototype from theoretical simulation and experimental verification to explore the formation of high PbSe(NaCl)_x_ (x = 3, 3.5, 4, 4.5) crystal quality and its inherent influence on TE properties. By the NaCl salt-assisted approach, the Na and Cl are involved in PbSe successfully, and the Na doping exerts a dominant influence on the valence band convergence. The charge carrier density of PbSe(NaCl)_x_ was gradually optimized from 2.71 × 10^19^ cm^−3^ to 4.16 × 10^19^ cm^−3^ as the increasing content of NaCl, concurrently with the appropriate Seebeck coefficient and conductivity value, a high power factor of 2.9 × 10^−3^ Wm^−1^K^−2^ and an appreciable ZT value of 1.26 at 690 K were caught by PbSe(NaCl)_4.5_. This demonstration indicates the positive role of Na^+^ and Cl^−^ on the thermoelectric performance of PbSe compounds, suggesting an effective strategy for upgrading the thermoelectric characteristics of thermoelectric materials by tuning the carrier concentration through NaCl salt-assisted approach, thus hoping to promote the development in PbSe-based TE technology for addressing global energy and environmental challenges. 

## 2. Results and Discussion

### 2.1. Structure and Composition 

The morphological image of PbSe(NaCl)_3_ with metallic luster and about 12 mm in size, shown in Figure 1a. Figure 1b demonstrates the X-ray diffraction (XRD) patterns of all samples; the standard XRD spectrum of PbSe is presented for a comparison at the bottom. It can be observed that the diffraction peaks of the doped sample were slightly shifted towards a high diffraction angle along with the introduction of NaCl, especially the peak at 60.381° in the inset. Generally, the shift of XRD peaks reflects residual stress or variations in chemical constituents [32]. In this context, the primary cause of the peak shift is the relatively reduced lattice constant, which can be attributed to the comparatively smaller ionic radius of Na^+^ (102 picometer) and Cl^−^ (181 picometer) in comparison to Pb^2+^ (119 picometer) and Se^2−^ (198 picometer), respectively. It demonstrates that the doping content has not reached the solid solubility limit of PbSe and the successful presence of NaCl in the lattice of PbSe. The intensity of the (200) diffraction peaks of PbSe(NaCl)_x_ enhanced and then decreased with the addition of more NaCl. At the same time, there is no discernible variation in other peaks, indicating that the Na^+^ and Cl^−^ facilitate the crystallinity, specifically the growth of (200) crystal planes. 

Figure 2a exhibits the FESEM pictures of the PbSe(NaCl)_3_ sample, without any cracks or voids, even with a magnification of 100,000 times. And the SEM analysis with different magnifications and multiple locations emerged in Appendix A, indicating the feasibility of the NaCl salt-assisted approach to synthesize the dense crystal structure, which agrees with the high density obtained from the Archimedes method. The range of this (8.47–8.74 g cm^−3^) is in close agreement with the theoretical density of 8.27 g cm^−3^ [33]. To further determine the elemental composition, an EDS map was performed on the PbSe(NaCl)_3_ in Figure 2b. Additionally, the results suggested that only Pb, Se, Na, and Cl elements were present in the sample without any other impurity elements. The electron probe microscope (EPMA) was used to investigate the content of every component, and the pertinent statistics are presented in Table 1. By increasing the proportion of NaCl, the content of Cl rose significantly relative to the content of Na, from 0.61% to 1.91%. The content difference of Se is similar, and the actual content of Pb in the sample is slightly reduced, indicating the substitute doping of NaCl inter to the lattice of PbSe. HRTEM was implemented to characterize the crystals at x = 3 (Figure 2c). The completed lattice stripes of PbSe(NaCl)_3.0_ display interplanar spacings of 0.306 nm in the particle, which matched well with the (200) planes of PbSe. Figure 2d exhibits the image of SAED acquired along the [002] zone axis of PbSe(NaCl)_3.0_, further confirming the high crystal quality of the specimen synthesized via the NaCl salt-assisted approach [34]. 

The XPS technology was employed to explore the chemical oxidation state of constituents in the sample, as displayed in Figure 3. The Pb 4f_7/2_ and Pb 4f_5/2_ orbital peaks of PbSe samples can be observed at 137.6 and 142.4 eV in Figure 3a, which corresponds to Pb^2+^ valence, indicating that the Pb element has a valence of +2. In Figure 3b, the XPS spectrum of Se 3d is presented, revealing the chemical binding energy peak of Se 3d_5/2_ at 53.4 eV and the chemical binding energy peak of Se 3d_3/2_ at 54.2 eV. These peaks correspond to the characteristics of Se^−2^, indicating that the Se element is −2 valence [34]. In Figure 3c, the XPS spectrum of Na 1s is displayed, and the chemical binding energy peak of Na 1s is observed at 1071.46 eV, suggesting that Na has a valence state of +1, thus promising the increase in hole concentration. Figure 3d exhibits the XPS spectrum of Cl 2p, where the chemical binding energy peak of Cl 2p is located within the range of 198.5–199 eV, implying that Cl has a valence state of −1.

The electronic structure of TE material can be commendably optimized by chemical doping, determining the carrier concentration and ultimately affecting the TE performance of material [35,36]. Therefore, it is crucial to explore the electronic structure by theoretical calculations. Based on the XRD and XPS results, it is not double that the Na^+^ and Cl^−^ ions are inserted in the PbSe’s lattice by substitutional doping. In order to show the way of NaCl doping intuitively and the influence of doping content on electronic structure of PbSe(NaCl)_x_ compounds, supercells with standards of 3 × 3 × 3 and 4 × 4 × 4 were built and further electronic structure information was explored. The construction of the PbSe (NaCl)_x_ model involves the substitution of Pb with Na and Se with Cl. We replaced Pb and Se with different stations, according to the principle of minimum energy to determine the optimal structure, the most stable configuration of the PbSe(NaCl)_x_ system is shown in Appendix A, which is utilized to further calculate the density of states (DOS) and band structure (Figure 4). Although these models cannot accurately correspond to the experimental doping concentration, they are sufficient to explain the effect of doping on the electronic structure and even properties of PbSe (NaCl)_x_ compounds. Figure 4a displays the energy band structure of the primary PbSe with a band gap of 0.419 eV, which corresponds well with the value of 0.439 eV calculated by J.P. Perdew et al. [37], and is also in good agreement with the bandgap range of 0.4–0.475 eV for PbSe thin films [38]. As shown in Figure 4b, it can be concluded that the L bands and the second valence band (Σ bands) can be found at the valence band regardless of doping NaCl, which is determined by the intrinsic properties of PbSe. In addition, the energy difference between the L and Σ bands is reduced from 0.28 eV in undoped PbSe to 0.18 eV in PbSe-NaCl, indicating a convergence of the two bands upon NaCl doping, of which is beneficial to enhance the TE performance of NaCl-doped PbSe. Similarly, this “band convergence” is also observed with the supercell of 4 × 4 × 4 (Appendix A), and the tendency of band structure is consistent with that of 3 × 3 × 3. Furthermore, in order to comprehensively explore the function of Na and Cl dopants in PbSe(NaCl)_x_ compounds, additional calculations were performed for Na and Cl doping in PbSe, as presented in Appendix A, respectively. The results indicate that both Na and Cl doping exhibit a significantly similar band convergence effect. Nonetheless, the ∆Ev of Na-doped PbSe is 0.18 eV, which is significantly smaller than that of Cl-doped PbSe (0.22 eV). This suggests that Na doping exerts a dominant influence on the valence band convergence, which is related to the report by Wu et al. [39], thereby leading ultimately to the improvement in electron transport and thermoelectric performance [17]. Compared with the calculated projected density-of-states of pure PbSe (Figure 4c), Na, Cl (Appendix A), or NaCl-doped PbSe (Figure 4d) systems, the conduction band minimum stems mainly from Pb p states, while the valence ∑ band comes from p orbitals of Se, Pb, and Cl for PbSe-NaCl.

### 2.2. Thermoelectric Properties

In Figure 5a–c, temperature-dependent electrical conductivity, the Seebeck coefficient, and a power factor of PbSe(NaCl)_x_ (x = 3, 3.5, 4, 4.5) are presented, respectively. The charge transport ability of thermoelectric materials can be judged by a conductivity test, and the result is shown in Figure 5a. With the temperature rising from 300 to 700 K, the electrical conductivity of PbSe(NaCl)_x_ compounds has a downward tendency, suggesting typical, degenerated semiconductor behavior. As the temperatures rise, the atoms and molecules in the material vibrate more violently, leading to an increase in the scattering of electrons in the material, thereby decreasing the conductivity. Notably, for temperatures beyond 600 K, PbSe-NaCl has a slower rate of decrease in conductivity since the suppression of bipolar conduction. The electrical conductivity of all samples is in the range from 12 × 10^4^ S m^−1^ to 18 × 10^4^ S m^−1^ at 300 K, with the most significant conductivity achieved for the sample defined as PbSe(NaCl)_4.5_. In combination with the Hall tests, further electrical properties of the samples were identified according to the calculated carrier concentration n_H_ (n_H_ = 1/(eR_H_)) and carrier mobility μ_H_ (μ_H_ = σR_H_) at room temperature (Table 1). As a result, the Hall coefficients *R_H_* of all samples are positive, indicating that hole conduction plays a dominant role in the samples. With an increase in the content of NaCl, the hole concentration increased gradually, resulting in a higher carrier concentration, as previously reported [40]. The carrier concentration increased from 2.71 × 10^19^ cm^−3^ to 4.16 × 10^19^ cm^−3^ because of the substitution of Na^+^ for Pb^2+^ and Cl^−^ for Se^2−^. Simultaneously, the carrier mobility decreased to 270 cm^2^ V^−1^s^−1^ in S4, which may be induced by the higher defect density. The carrier mobility and concentration together determine the conductivity of the semiconductor according to σ = neμ. The results reveal that the higher the content of NaCl, the better the conductivity; the maximum electrical conductivity of 18 × 10^4^ S m^−1^ was achieved in PbSe(NaCl)_4.5_, indicating that NaCl can be used as an excellent dopant for transferring electronics, in agreement with the conclusion reported by Jing Li et al. [41].

The Seebeck coefficient refers to the ratio between the voltage generated by a thermoelectric material under a temperature gradient and the temperature gradient, which is one of the key parameters to measure the performance of TE materials. It is obvious that the Seebeck coefficients of samples are positive (Figure 5b), indicating that the PbSe(NaCl)_x_ samples are p-type semiconductors, and the values of S increase with the rising temperature. It is noteworthy that the Seebeck coefficient value of PbSe(NaCl)_x_ is about 66~80 mV/K, which is two quantities larger than that of pure PbSe compounds prepared by the conventional melting technique or the microwave-assisted chemical deposition (MA-CBD) technique [42]. This improvement corresponds well to the energy band convergence of PbSe(NaCl)_x_. The relationship between power factor (PF), calculated according to PF = *a^2^σ,* and temperature is shown in Figure 5c. At the outset, the PF increases slightly from 7 × 10^−4^ W m^−1^K^−2^ to 8 × 10^−4^ W m^−1^K^−2^ with the increasing content of NaCl at room temperature, and the PF increases with the increasing temperature for all samples. As a consequence of the higher *σ* combined with the excellent *a*, the maximum power factor of 2.9 × 10^−3^ W m^−1^ K^−2^ was achieved in PbSe(NaCl)_4.5_ at 700 K, which exceeds the value obtained for the sample prepared by the melt rotation method (1.25 × 10^−3^ W m^−1^K^−2^) [43]. 

In addition, thermal conductivity is also one of the important factors affecting the properties of thermoelectric materials, which refers to the ability of heat transport in a material. Figure 5d,e show the temperature-dependent thermal conductivity (total (κ) and lattice (κ_L_)) of samples with varying content. In thermoelectric materials, thermal conductivity is influenced by the lattice structure and the transport model charge. The total thermal conductivity at room temperature of PbSe(NaCl)_3.0_ is ~2.54 W m^−1^ K^−1^, rising to ~3.0 W m^−1^ K^−1^ for PbSe(NaCl)_4.5_, which may be due to the increases in electronic thermal conductivity caused by the higher doping concentration. Over the temperature span of 300 K to 700 K, the total thermal conductivity of all specimens decreased monotonically with the escalating temperature. The main reason for this is the increase in the intensity of phonons vibration with the rising temperature, leading to more frequent scattering and a decrease in phonon thermal conductivity, thus resulting in a decrease in the total thermal conductivity. Additionally, after 500 K, the thermal conductivity decreases more slowly with temperature due to the intensification of lattice vibration and the appearance of a bipolar effect [34]. 

The lattice thermal conductivity (κ_L_) was estimated by subtracting the electronic thermal conductivity from the total thermal conductivity (κ_L_ = κ − κ_e_). The electronic thermal conductivity was calculated according to the Wiedemann–Franz law, κ_e_ = LσT, where L is the Lorenz number (L = 2.45 × 10^−8^ W Ω K^−2^) [44]. At room temperature, the lattice thermal conductivity *κ_L_* of all PbSe-(NaCl)_x_ samples are in the range of 1.7 ± 0.2 W m^−1^ K^−1^, consistent with the previous report [31,45], and that of PbSe(NaCl)_3.0_ is around 1.67 W m^−1^ k^−1^. With the increasing temperature, the lattice thermal conductivity of PbSe(NaCl)_3.0_ shows a downward trend to the lowest value of ~0.99 W m^−1^ K^−1^ at 630 K contributed by the lattice vibration. Surprisingly, it further decreased to ~0.72 W m^−1^K^−1^ at ~660 K as the stoichiometric ratio of NaCl increased to 4.5, which was primarily caused by the enhancement of the point defect scattering from the high doping level [43,46]. The point defect scattering in solid solution systems originates from the differences in atomic mass (mass fluctuations) and size, as well as the interatomic coupling force differences (strain fluctuations) between the impurity atoms and the host lattice, as described by the Callaway model [47,48]. Here, the substantial atomic mass fluctuation is mainly manifested between Na (atomic mass is 23) and Pb (atomic mass is 207), which predominately contributes to the reduction in lattice thermal conductivity. In particular, the lattice thermal conductivity of PbSe(NaCl)_4.5_ is slightly increased at 690 K, which may be due to the redistribution of doping elements at this temperature, resulting in changes in the lattice structure, and then changing the propagation mode of phonons, engendering a slight increase in the κ_L_. 

Importantly, In Figure 5f, the temperature-dependent ZT is presented, which is the most convincing, and highlights the impressive thermoelectric performance of heavily doped p-type PbSe. At a temperature of 690 K, a ZT value of ~1.26 was observed when the carrier concentration, n_H_, was 4.2 × 10^19^ cm^−3^. The figure is superior to that of PbSe-based materials processed using the hot-pressing technique (~1.12 at 690 K) and spark plasma sintering (~1.0 at 690 K), indicating that the low thermal conductivity and significantly improved power factor (PF) in heavily doped samples contribute to the promising thermoelectric performance in p-type PbSe. The reduction in thermal conductivity can be attributed to the introduction of dopants, which scatter phonons and decrease their mean free path. Meanwhile, the high carrier concentration provokes a remarkable amelioration in electrical conductivity and power factor. These two factors, in combination, result in a high ZT value for the heavily doped p-type PbSe material. The promising thermoelectric performance of heavily doped p-type PbSe suggests its potential use in various applications, such as waste heat recovery, thermoelectric cooling, and power generation.

## 3. Materials and Methods

### 3.1. Synthesis Method

Stoichiometric quantities of high-quality raw materials, namely, 99.99% lead (Pb) shots, 99.999% selenium (Se) shots, and 99.999% sodium chloride (NaCl) granules, all purchased from Aladdin, were precisely measured in the proportion of 1:1:x (where x = 3, 3.5, 4, or 4.5) and labeled as S1, S2, S3, and S4 in sequence. The raw materials were mixed thoroughly using a high-energy ball mill (model XQM-4L, Tencan Powder Technology Co., Ltd. Changsha, China) for 4 h under an argon atmosphere. The mixture was then loaded into a vitreous silica tube (diameter = 30 mm, length = 500 mm) with a carbon layer, which was subsequently evacuated and sealed under a vacuum. The sealed tube was placed in a horizontal tube furnace (model STF-1200, Kejing Electric Furnace Co., Ltd., Foshan, China) and heated to 932 K at a rate of 178 K/h, held at this temperature for 3 h, and then heated to 1373 K for 1 h, and kept at this temperature for 20 h. After that, the temperature was decreased to 723 K within 60 h and then cooled to room temperature. Distilled water was used to extract the lustrous crystals from the solvent, and the samples were tested with a series of representations.

### 3.2. Characterization and Measurement

The crystal structure of the sample was determined through powder X-ray diffraction (XRD) analysis using Cu Kα radiation at an ambient temperature, employing an Ultima IV instrument. Electron probe microanalysis (EPMA) (JXA-8230) was utilized to examine the actual contents of each element in the samples. Field emission scanning electron microscopy (FESEM) (SUPRA 55VP) was utilized to analyze the microstructures of the samples, and qualitative elemental analysis of the samples was conducted using electron energy dispersive X-ray spectroscopy (EDS) with an Oxford instrument. Chemical analysis of Pb, Se, Na, and Cl chemical states was performed using X-ray photoelectron spectroscopy (XPS) on a PHI5000 Versaprobe-II system. Furthermore, the transmission electron microscope (TEM) was used to obtain images, along with corresponding selected area electron diffraction (SAED) patterns, using a Tecnai G20 instrument. The Seebeck coefficient (α) and electrical conductivity (σ) were assessed from 300 K to 690 K in the vacuum condition. The common four-point probe approach measured the σ with a direct current of 20 mA. The α were attained, employing a comparison approach with Konstantin (Ni:40%) as a guide and a temperature gradient of approximately 1.5 K. We usually use the formula α = ΔV/ΔT to calculate the Seebeck coefficient, where ΔV is the thermoelectromotive force, and ΔT is the temperature difference between the two sides of the material. The thermal conductivity (κ = DC_P_*ρ*) was directly obtained by the transient plate heat source method, in which the density *ρ* was measured by the Archimedes method. Electronic thermal conductivity κ_e_ and lattice thermal conductivity κ_L_ were calculated according to κ_e_ = LσT and κ_L_ = κ_total_ − κ_e_, respectively. The Hall coefficient (R_H_) was determined using the van der Pauw method and the HL5500 Hall system from Nanometrics, under a magnetic field of 0.55 T and a direct current of 1 mA at room temperature. The samples, which had thicknesses ranging from 0.3 to 0.5 mm, were subjected to this analysis. The Hall carrier concentration n_H_ and Hall mobility μ_H_ were determined using the formulas *n* = 1/(eR_H_) and μ_H_ = σR_H_, respectively.

### 3.3. Details of Computational Methods

The projector augmented wave (PAW) method within the density functional theory (DFT) was employed using the Vienna ab initio simulation package (VASP) for all computations. The Perdew–Burke–Ernzerhof (PBE) functional and an energy cutoff of 400 eV were utilized for the density-of-states and band-structure calculations based on a 3 × 3 × 3 supercell. The energy and relaxation force convergence criteria were set to 10^−5^ eV and 0.01 eV Å^−1^, respectively. For the crystal structure relaxation, a k-grid of 3 × 3 × 3 was used, while a k-grid of 5 × 5 × 5 was employed for the static calculation in the intrinsic supercell and PbSe(NaCl)_x_. The density-of-states calculation utilized a k-grid of 7 × 7 × 7. In addition, the band structure of pristine PbSe and PbSe-NaCl was calculated by considering spin–orbit coupling (SOC).

## 4. Conclusions

In summary, the influence of different levels of NaCl doping on the microstructure, energy band structure, and thermoelectric properties of PbSe-(NaCl)_x_ are systematically investigated by the first-principles calculations combined with practical experiments. All samples exhibited the PbSe-type crystal structure—without any other phase. Based on the NaCl salt-assisted approach, the carrier concentration of PbSe(NaCl)_x_ is regulatable by the doping content of NaCl, which was adjusted up to 4.2 × 10^19^ cm^−3^ in PbSe(NaCl)_4.5_, leading to the appreciable α and σ simultaneously, thereby resulting in a super ZT value (1.26 at 690 K) eventually. The feasibility of the NaCl salt-assisted approach to modify the TE performance of the PbSe compound promotes the development of PbSe-based TE technology for addressing global energy and environmental challenges and providing reference for other TE materials. 

## Figures and Tables

**Figure 1 molecules-28-02629-f001:**
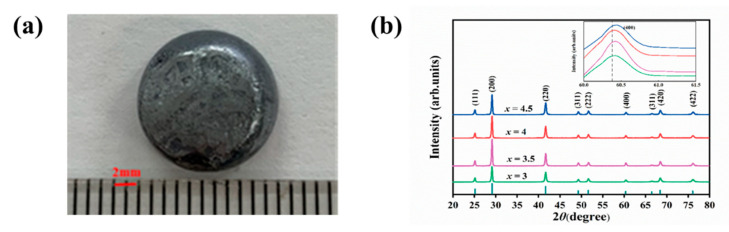
External appearance picture and X-ray diffraction patterns. (**a**) The photograph of PbSe(NaCl)_3_ synthesized using the sodium chloride fusion approach. (**b**) XRD patterns of PbSe(NaCl)_x_ (x = 3, 3.5, 4, 4.5) with the standard XRD spectrum of PbSe for comparison.

**Figure 2 molecules-28-02629-f002:**
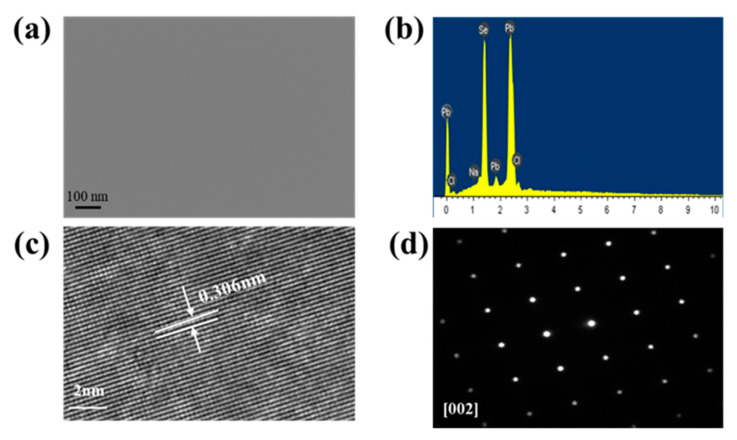
The micro characterization of PbSe(NaCl)_3_ with high crystal quality. (**a**) FESEM image, (**b**) EDS image, (**c**) HRTEM image, (**d**) SAED image.

**Figure 3 molecules-28-02629-f003:**
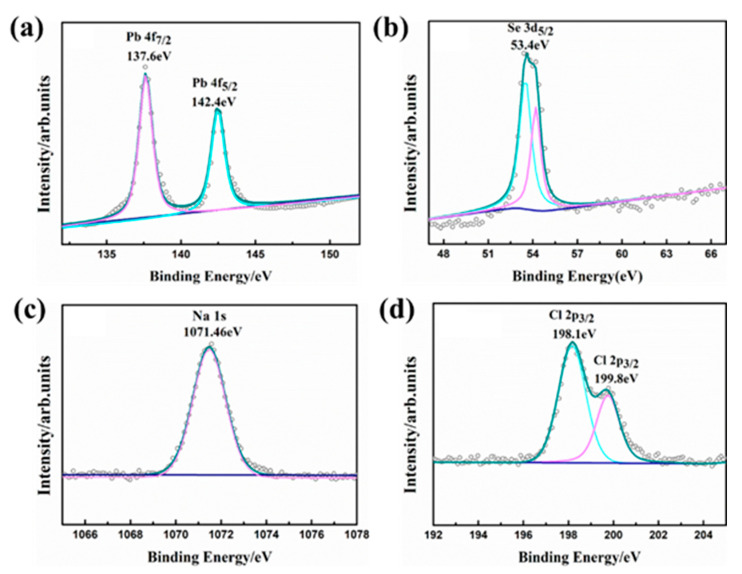
XPS fittings for Pb 4f (**a**), Se 3d (**b**), Na 1s (**c**), and Cl 2p (**d**).

**Figure 4 molecules-28-02629-f004:**
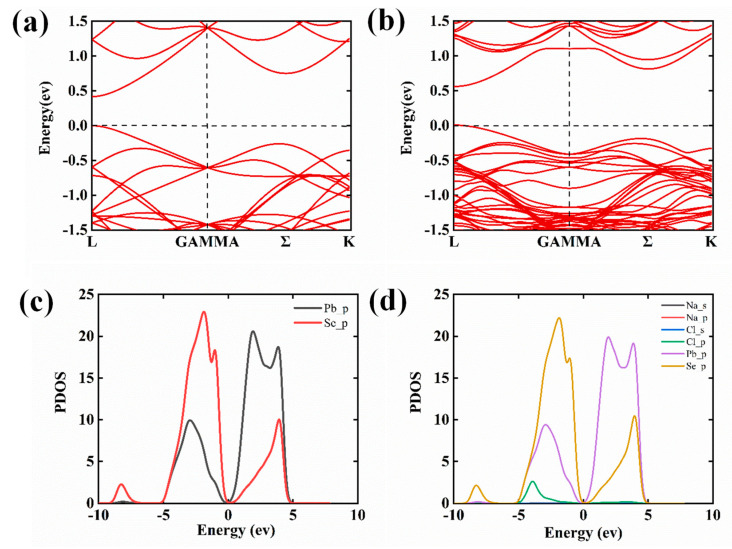
Band structure of pure PbSe with ∆Ev~0.28 eV (**a**); NaCl-doped PbSe with ∆Ev~0.18 eV (**b**); the corresponding DOS for pure PbSe (**c**) and PbSe with NaCl doping (**d**), where the corresponding states are presented in a clear manner to demonstrate their respective contributions.

**Figure 5 molecules-28-02629-f005:**
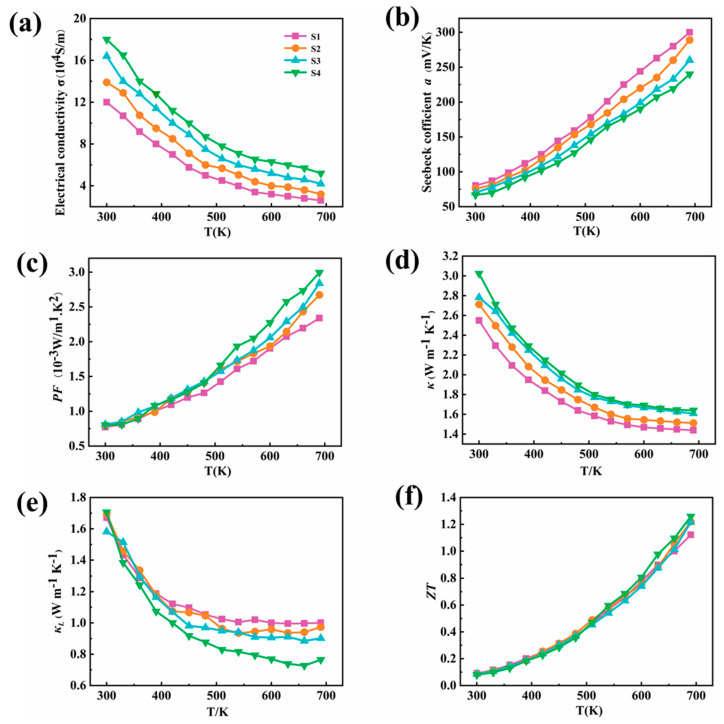
Temperature-dependent (**a**) electrical conductivity, (**b**) the Seebeck coefficient, (**c**) power factor, (**d**) thermal conductivity, (**e**) lattice thermal conductivity and (**f**) figureof merit ZT for PbSe(NaCl)_x_ (x = 3, 3.5, 4, and 4.5).

**Table 1 molecules-28-02629-t001:** A list of samples and some of their temperature-dependent properties at 300 K.

Samples	Crystal Compositions				*R_H_*	*n_H_*	*μ_H_*	*ρ*
	PbSe-(NaCl)*x*				cm^3^/C	10^19^ cm^−3^	cm^2^/Vs	g·cm^3^
	Pb	Se	Na	Cl				
*x* = 3	48.52	50.7	0.17	0.61	0.23	2.717	276	8.47
*x* = 3.5	48.17	50.6	0.2	1.03	0.21	2.976	292	8.59
*x* = 4	47.96	50.1	0.3	1.64	0.18	3.472	295	8.52
*x* = 4.5	47.58	50.1	0.41	1.91	0.15	4.167	270	8.74

## Data Availability

Request for the data presented in the study can be made to the corresponding author.

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
