# Peer review of "Preparation of Heavily Doped P-Type PbSe with High Thermoelectric Performance by the NaCl Salt-Assisted Approach"

_molecules, 2023, doi:10.3390/molecules28062629_

Round 1

Reviewer 1 Report

This work deals with the influence of different levels of NaCl doping on the microstructure, energy band structure and thermoelectric properties of PbSe. With the successful introduction of NaCl, both theoretical and experimental results demonstrate that manipulating the carrier concentration by adjusting the content of NaCl is conducive to upgrading the electrical transport properties of the materials. At the same time, NaCl as a p-type dopant provides holes for increasing carrier concentration in PbSe(NaCl)4.5 with a positive Seebeck coefficient, contributing to an enhanced power factor. As a consequence, a high ZT value (1.26 at 690 K) was realized in PbSe(NaCl)4.5, which is more optimistic in comparison with that of other PbSe-based materials. 

Author Response

Thanks for the reviewer's comments.

Reviewer 2 Report

In this manuscript, the authors prepared of heavily doped P-type PbSe with high thermoelectric performance by NaCl-Flux Method. The research work represents scientific-interesting. The manuscript can be improved if the authors can revise the following points:

1.     Multiple citations for single point should be avoided; such as [7-9], [10-12].

2.     Please raise the readability of this manuscript by a substantial explanation of the results and discussion.

3.     The authors came to the conclusion that a high thermoelectric figure of merit (ZT) of 1.26 at 690 K indicates that temperature is significant. Why does the author not raise the temperature more?

Author Response

We appreciate the constructive suggestions of the reviewer and we have revised the manuscript following his suggestions. The changes to the main text are highlighted in yellow in the manuscript document. Below we reply to his specific queries.

Q2.1: Multiple citations for single point should be avoided; such as [7-9], [10-12].

Response: Thanks for pointing out the above error, and it has been corrected in the revised manuscript.

Q2.2: Please raise the readability of this manuscript by a substantial explanation of the results and discussion.

Response: Thanks for the valuable comment. As suggested, reasonable and appropriate explanation have been provided in the results and discussion section, and the corresponding analysis has been shown in the revised manuscript.

Such as on page 9 line 23 add sentence ‘In order to comprehensively explore the function of Na and Cl dopant in PbSe(NaCl)x compounds, additional calculations were performed for Na and Cl doping in PbSe, as presented in Figure S2 (a) and (c), respectively. The results indicate that both Na and Cl doping exhibit a significantly similar band convergence effect. Nonetheless, the ∆Ev of Na doped PbSe is 0.18 eV, which is significantly smaller than that of Cl doped PbSe (0.22 eV). This suggests that that Na doping exerts a dominant influence on the valence band convergence, which is related to the the report by Wu et al [34], thereby leading to improvement of electron transport and thermoelectric performance ultimately [17].’ to further explain NaCl doping can affect the electronic structure, which is beneficial to improve the electrical transmission performance of thermoelectric materials.

On page 12 line 4 add sentence ‘The main reason for this is resulting from the increases intensity of phonons vibration with the temperature rise, leading the scatter more frequently and decreasing the phonon thermal conductivity, resulting in a decrease in the total thermal conductivity.’ to further explain over the temperature span of 300 K to 700 K, the total thermal conductivity of all specimens decreased monotonically with escalating temperature.

Q2.3: The authors came to the conclusion that a high thermoelectric figure of merit (ZT) of 1.26 at 690 K indicates that temperature is significant. Why does the author not raise the temperature more?

Response: Thanks for the valuable comment.

As the PbSe is a TE material suitable for medium and low temperature applications, the temperature range is 300-700 K, and due to the equipment limitation, the test temperature is to 690 K. Most importantly, the chemical doping effects on thermoelectric performance of PbSe-(NaCl)x has been well reflected in this temperature range.

Reviewer 3 Report

The authors study by means of experiments and firs-principles simulations the thermoelectric properties of PbSe compounds. In particular, they consider the effect of doping with NaCl and find that it facilitates the cristalinity and enhances the power factor by increasing the carrier concentration. They conclude that this way of chemical doping increases the thermoelectric performance.

I find the study scientifically sound and interesting. The article is well written, the figures are clear and the explanations seem to be correct. I think the article could be further considered after the authors address the following comments:

- The supercell used in the simulations is not large enough to account for different ranges of concentrations, allowing only relatively large values of such concentrations. The authors should elaborate and comment on that on the text.

- It would be good to give more details in the main text, such as the way the Na and Cl atoms are included to add the doping in the simulations.

- The DFT gap has not been corrected. Are the results with such gap valid?

- It is not clear what the use of the bands and the DOS can be. Why are such properties relevant, apart from showing the opening of the gap? It would be much more helpful the calculation of the thermoelectric properties from such results and their comparison with the experiments. Can the authors do such calculations or, at least, comment on that?

Author Response

We appreciate the constructive suggestions of the reviewer and we have revised the manuscript following his suggestions. The changes to the main text are highlighted in yellow in the manuscript document. Below we reply to his specific queries.

Q4.1: The supercell used in the simulations is not large enough to account for different ranges of concentrations, allowing only relatively large values of such concentrations. The authors should elaborate and comment on that on the text.

Response: Thanks for the valuable comment. As mentioned in the Computational details, the Perdew-Burke-Ernzerhof (PBE) functional and energy cutoff of 400 eV were taken for density of states and band structure calculation based on a 3×3×3 supercell has been employed to predict the effects of Na+ and Cl- doping on the structural optimization or electronic band regulation of PbSe(NaCl)x compounds, which would optimize the carrier concentration or TE performance ultimately, it is difficult and unnecessary to control the doping concentration precisely, and this method is wildly used in the research on the chemical doping effects of alloy compounds, such as Kim, H.S. et al. investigated the thermoelectric transport properties of Pb-doped SnSe alloys (PbxSn1-xSe) by establishing 1×2×2 supercells (The citation is 34 of the references).

Q4.2: It would be good to give more details in the main text, such as the way the Na and Cl atoms are included to add the doping in the simulations.

Response: Thanks for the valuable comment. As suggested, added sentence ‘Based on to the XRD and XPS results, it is no double that the Na+ and Cl- ions are inserted in the PbSe’s lattice by substitutional doping. In order to show the way of NaCl doping intuitively, a 3×3×3 supercell has been built to show the atomic station in PbSe(NaCl)x. The construction of the PbSe (NaCl)x model involves the substitution of Pb with Na and Se with Cl. Replace Pb and Se with different stations, according to the principle of minimum energy to determine the optimal structure, the most stable configuration of the PbSe(NaCl)x system was shown in Figure S1,which is utilized to further calculating the DOS and band structure.’ on page 9 line 6 to enrich the details of calculation.

Q4.3: The DFT gap has not been corrected. Are the results with such gap valid?

Response: Thank you for your feedback. In this work, the electrical structure of PbSe(NaCl)x compounds are calculated by PBE method, and the band gap of the original PbSe is 0.419 eV, which is corresponding well with the value of 0.439 eV calculated by J.P. Perdew et al. ( Perdew et al. Physical review. B, Condensed matter 1996, 54: 16533-16539, and which is cited as reference [35]), the corresponding describes can be found in the revised manuscript on page 9 line 15. Additionally, this method is popular used to calculate the gap of chalcogenide compounds, indicating the reliability of the calculation results.

Q4.4: It is not clear what the use of the bands and the DOS can be. Why are such properties relevant, apart from showing the opening of the gap? It would be much more helpful the calculation of the thermoelectric properties from such results and their comparison with the experiments. Can the authors do such calculations or, at least, comment on that?

Response: Thank you for your feedback. The use of energy bands and density of states not only provides important information about the electronic structure, but also can be used to predict the electrical transport properties of materials. In addition, the band structure can provide insight into the degree of band convergence or band degeneracy, which has a direct impact on the thermoelectric properties of the material. It can be found that the NaCl dopant incorporation leads to band convergence and the S band is involved in electron transport, leading the increase of power factor. Therefore, the band structure and DOS play an important role to understand the enhancement mechanism of thermoelectric performance about chemical doping in this study, the corresponding describes can be found in the revised manuscript on page 9 line 21.

In this work, we have primarily focused on the experimental measurements of the thermoelectric properties, and have correlated those with the theoretical results. We have discussed the effect of NaCl doping on the electronic transport properties, which directly affects the TE performance of the material. Of course, we also possessed the thermoelectric properties of materials by theoretical calculation in our previous work (references [33]), and welcome to follow.

Reviewer 4 Report

Please see the comments file.

Author Response

We appreciate the constructive suggestions of the reviewer and we have revised the manuscript following his suggestions. The changes to the main text are highlighted in yellow in the manuscript document. Below we reply to his specific queries.

Q3.1: The keywords should be arranged in alphabetical order, and the authors should pick the most relevant and understandable keywords specific to the present research work. Add more keyword, should not be less than six.

Response: Thanks for the valuable comment. The keyword has been revised and added as ‘Chemical doping; Salt-assisted; NaCl doping; PbSe; Thermoelectric material; Thermoelectric performance’, which has been corrected in the revised manuscript and highlighted in yellow.

Q3.2: The opening paragraph of the “introduction section” is very weak and lacks the elaboration of the problem statement. Why the thermoelectric materials are the only solution for direct conversion of the scrap heat, while there are many other emerging technologies efficient than the thermoelectric materials.

Response: As suggested, the “introduction section” has been systematically improved in the revised manuscript and highlighted in yellow. The opening paragraph of the “introduction section” has been modified as ‘Thermoelectric (TE) materials capable of transforming scrap thermal energy into electric power, have attracted considerable attention [1,2]. To promote the sustainable development of the environment and economy, it is crucial to explore suitable TE materials with high TE performance for their practical application. Currently, based on the existing thermoelectric materials such as Bi2Te3 and SiGe, the corresponding conversion efficiency of them are around with the value of 10%, which is far away from the demanding value of 20% [3]. Among the advanced thermoelectric materials, lead chalcogenide has generated widespread interest due to its low thermal conductivity, high carrier fluidity, and high symmetrical structure [4,5], in which the PbTe has been widely studied. Notably, to the high melting point and rich content of the element, the PbSe (a close analog of PbTe), which with a similar two-valley valence band structure to PbTe has also captivated extensive research, but the unsatisfactory TE properties still hinder its practical application [6]. To obtain higher conversion efficiency, it is necessary to improve the dimensionless figure of merit (ZT), which is described as ZT=a2s T⁄k, Where σ represents the electrical conductivity and α denotes the Seebeck coefficient. The product a2σ is referred to as the power factor. The total thermal conductivity is composed of the lattice thermal conductivity () [7,8] and the electron thermal conductivity (), and T represents the absolute temperature. Generally, there are two ways to enhance the ZT values, one is optimizing carrier concentration and changing the energy band to boost the PF, and the additional choice is reducing the lattice thermal conductivity through the initiation of atomic defects and nanostructures [9-12]. Excitingly, chemical doping can not only modify the electronic structure but also optimize the carrier concentration, which has been illustrated as a successful control approach for tuning the TE material properties [13-15].’.

Thermoelectric materials capable of converting scrap thermal energy into electric power, has attracted considerable attention. In order to promote the sustainable development of environmental and economic, it is crucial to explore the suitable TE materials which with high TE performance for their practical application.

Q3.3: The rest of the "introduction section" is also very weak. The authors have cited multiple research studies; however, the literature's critical contribution and outcomes need more elaboration. There is also a dire need to explore the critical parameters that have not been considered in the cited literature and development of a statement of the novelty of the present work in the context of reported studies.

Response: Thanks for your feedback. The “introduction section” has been systematically improved in the revised manuscript. Such as the sentence on page 3 line 22 ‘Excitingly, chemical doping can not only modify the electronic structure but also optimize the carrier concentration, which has been demonstrated as an effective control strategy for tuning the TE material properties.’ and on page 4 line 18 ‘Above all, chemical doping combined with the suitable preparation process, the PbSe-based materials which with high TE performance could be achieved by structural optimization or electronic band regulation [29], and further research is urgently needed to advance the development of PbSe-based TE technology.’ has been added to expound the literature's critical contribution and reveal the novelty of the present work.

Q3.4: The authors did not present their objectives clearly in the last paragraph of the “introduction” and the research novelty, based on the aforementioned literature. The authors are desired to add a paragraph elaborating comprehensively the objectives to describe in detail the significance, novelity and the core purpose of this research study.

Response: Thanks for the valuable comment. In the revised manuscript, the sentence ‘Here we present tuning TE performance via chemical doping and process improvement by NaCl flux, employing the Na and Cl substitution as prototype from theoretical simulation and experimental verification to explore the formation of high PbSe(NaCl)x (x = 3, 3.5, 4, 4.5) crystal quality and its inherent influence on TE properties.’ on page 4 line 22 and ‘which suggests an effective strategy for upgrading thermoelectric characteristics of thermoelectric materials by tuning carrier concentration.’ on page 4 line 4 have been added in the revised manuscript of the “introduction” to present the purpose and novelty.

Q3.5: Authros should expand the materials and methods section and shift it to before results and discussion. Describe about the nature of materisls used, their specifications, the experimental setup for the execuation of the research study, the specifications of the instrumets used and uncertanity and the procefures adapted in detail.

Response: Thanks for your feedback. We have moved the Materials and Methods section before the Results and Discussion part. In addition, we have modified this section accordingly and highlight in yellow. Hopping these changes will improve the clarity and transparency of our research.

Q3.6: The authors mentioned that they have found no cracks after SEM analysis at a magnification of 6000, however, it is proposed to perform the SEM analysis from multiple locations up to 100000 magification to strengthen the statement.

Response: Thanks for the valuable comment. As suggested, the previous SEM detection has been replaced by the image with higher magnification in Figure 2 (a), and it was found no cracks in SEM analysis from multiple locations.

Q3.7: In the whole results and discussion, section authors have mostly described the Figures and the critical discussion of the findings is missing. The authors are desired to present a critical discussion on the findings presented in each figure by correlating with the existing literature. The novelty of the present work is still not much strong, and authors should focus on it by consulting more literature.

Response: Thank you for your valuable feedback. As suggested, comparations with other research has been supplemented, which can be find in Section 3. Results and discussion.

Added sentence on page 9 line 3, 'As the electronic structure of TE material can be commendably optimized by chemical doping, determining the carrier concentration and ultimately affecting the TE performance of material [33,34]. Therefore, it is crucial to explore the electronic structure by theoretical calculation.' to further discuss the necessity of band calculation.

On page 9 line 23, sentence ‘In order to comprehensively explore the function of Na and Cl dopant in PbSe(NaCl)x compounds, additional calculations were performed for Na and Cl doping in PbSe, as presented in Figure S2 (a) and (c), respectively. The results indicate that both Na and Cl doping exhibit a significantly similar band convergence effect. Nonetheless, the ∆Ev of Na doped PbSe is 0.18 eV, which is significantly smaller than that of Cl doped PbSe (0.22 eV). This suggests that that Na doping exerts a dominant influence on the valence band convergence, which is related to the the report by Wu et al [36], thereby leading to improvement of electron transport and thermoelectric performance ultimately [17].' has been supplemented to further discuss NaCl doping can affect the electronic structure, which is beneficial to improve the electrical transmission performance of thermoelectric materials.

Q3.8: The conclusion section looks more to reflect the study's results, and the conclusive image of the research needs to be further elaborated without further expanding this section. Also, the multiple paragraphs in the conclusion section should be avoided.

The overall content of the manuscript needs extensive improvements in terms of novelty and elaborations of the results and discussion in correlation with the existing studies. The manuscript has major grammatical errors in several places. Therefore, the authors are also desired to proofread the manuscript by an expert to avoid such major errors.

Response: Thanks for your valuable comments. The manuscript has been systematically revised and the modified part has been highlighted in yellow. We adjusted the paragraph’s structure of the introduction section, added sentence such as ‘Here we present tuning TE performance via chemical doping and process improvement by NaCl flux, employing the Na and Cl substitution as prototype from theoretical simulation and experimental verification to explore the formation of high PbSe(NaCl)x (x = 3, 3.5, 4, 4.5) crystal quality and its inherent influence on TE properties.’ on page 4 line 22 to enhance the logicality and readability of the article.

Round 2

Reviewer 3 Report

The authors have not properly replied to most of the comments. In particular:

- They say “it is difficult and unnecessary to control the doping concentration precisely…” This must refer to experiments and has no relationship to the theoretical calculations. Using larger supercells to change the doping has surely in most cases an important effect. The authors should clearly address this point in the text of the article.

- The gap should not be compared to previous simulations, that might suffer the same shortcomings given by DFT (50 % underestimation), but to experiments. The authors should give this comparison and elaborate in the main text why they think the DFT results would be valid. Also, the reference to page 9 is wrong, there is no such a line in page 9, be careful when you cite the text.

- The authors should also explain in the text why the use of the band structure and DOS is enough to derive the transport properties, how such transport properties are deducted from them and why it is not necessary to perform a transport simulation. Again, the reference to page 9 is wrong.

These issues should be clearly addressed in the following reply, otherwise I will not be able to accept the manuscript.

Reviewer 4 Report

Refer to the comments file
